# 2D-Hexagonal Boron Nitride Screen-Printed Bulk-Modified Electrochemical Platforms Explored towards Oxygen Reduction Reactions

**DOI:** 10.3390/s22093330

**Published:** 2022-04-26

**Authors:** Aamar F. Khan, Alejandro Garcia-Miranda Ferrari, Jack P. Hughes, Graham C. Smith, Craig E. Banks, Samuel J. Rowley-Neale

**Affiliations:** 1Faculty of Science and Engineering, Manchester Metropolitan University, Chester Street, Manchester M1 5GD, UK; aamar442@gmail.com (A.F.K.); a.garcia-miranda.ferrari@mmu.ac.uk (A.G.-M.F.); jackhughes723@gmail.com (J.P.H.); c.banks@mmu.ac.uk (C.E.B.); 2Manchester Fuel Cell Innovation Centre, Manchester Metropolitan University, Chester Street, Manchester M1 5GD, UK; 3Department of Natural Sciences, Faculty of Science and Engineering, University of Chester, Thornton Science Park, Pool Lane, Ince, Chester CH2 4NU, UK; graham.smith@chester.ac.uk

**Keywords:** boron nitride, screen-printed electrodes (SPEs), electrochemistry, oxygen reduction reaction (ORR)

## Abstract

A low-cost, scalable and reproducible approach for the mass production of screen-printed electrode (SPE) platforms that have varying percentage mass incorporations of 2D hexagonal boron nitride (2D-hBN) (2D-hBN/SPEs) is demonstrated herein. These novel 2D-hBN/SPEs are explored as a potential metal-free electrocatalysts towards oxygen reduction reactions (ORRs) within acidic media where their performance is evaluated. A 5% mass incorporation of 2D-hBN into the SPEs resulted in the most beneficial ORR catalysis, reducing the ORR onset potential by ca. 200 mV in comparison to bare/unmodified SPEs. Furthermore, an increase in the achievable current of 83% is also exhibited upon the utilisation of a 2D-hBN/SPE in comparison to its unmodified equivalent. The screen-printed fabrication approach replaces the less-reproducible and time-consuming drop-casting technique of 2D-hBN and provides an alternative approach for the large-scale manufacture of novel electrode platforms that can be utilised in a variety of applications.

## 1. Introduction

Electrochemical oxygen reduction reaction (ORR) in a hydrogen fuel cell is the most notable reaction within the field of energy generation. ORR can be applied to a diverse range of fields and electrochemical applications (such as wastewater management) [1]. Within an electrochemical system, ORR has a large positive theoretical thermodynamic cell potential of +1.23 V (vs. RHE), shown in Equation (1). This large onset potential is due to the significant kinetic inhibition associated with the cleavage of the (di)oxygen double bond [2,3]. In order to mitigate the over-potential of ORR, it is essential that efficient electrocatalytic materials are used. Typically, precious metal oxides, such as RuO_2_ and IrO_2_, are utilised in order to efficiently catalyse ORR. This is due to their negligible binding energies for O_2_ adsorbates; however, the application of such materials within commercial devices is undesirable due to their high cost and low earth abundancy [4,5,6,7]. Consequently, research has been directed towards the development of cost-effective nonprecious metal (NPM) ORR electrocatalysts [5,8]. The ORR mechanism can occur via two routes (acidic electrolyte or under standard conditions) dependent upon the catalytic material used [3,9].

A direct four-electron pathway is described as follows.
(1)$$O_{2}\;\left(g\right)+4H^{+}\;\left(aq\right)+4e^{-}\;\to 2H_{2}O\;{E^{o}}^{\;}=+1.23\;V\;(\mathrm{vs}.\mathrm{RHE})$$

Alternatively, a two-step, two-electron peroxide pathway is described as follows.
(2)$$O_{2}\;\left(g\right)+2H^{+}\;\left(aq\right)+2e^{-}\to H_{2}O_{2}\left(g\right)\;{E^{o}}^{\;}=+0.70\;V\;(\mathrm{vs}.\;\mathrm{RHE})$$
(3)$$H_{2}O_{2}\;\left(g\right)+2H^{+}\;\left(aq\right)+2e^{-}\to +2\;H_{2}O\;\left(l\right)\;{E^{o}}^{\;}=+1.76\;V\;(\mathrm{vs}.\;\mathrm{RHE})$$

Whilst much of the corresponding research has been focused on ensuring that ORR occurs via the four-electron ORR [10,11], materials that allow for the two-electron ORR pathway are also of great interest. For example, the reliable electrochemical synthesis of H_2_O_2_ is highly desirable within wastewater treatment applications that require a continuous supply of strong oxidising agents [1,12,13]. As a strong oxidant, H_2_O_2_ can oxidise organic pollutants found within waste water, via the Fenton reactions, into oxygen and water byproducts. Thus, the in situ electrochemical production of H_2_O_2_ can potentially remove environment damaging water pollutants, whilst also eliminating the costs and hazards associated with the production, transportation and handling of commercially available H_2_O_2_ [1].

There is a plethora of studies that report on NPM ORR catalysts with a great deal of attention being focused on certain two-dimensional (2D) materials, such as 2D-MoS_2_ [3], 2D-MoSe_2_ [14,15] and 2D-WSe_2_ [14,15]. Two-dimensional hexagonal boron nitride (2D-hBN) is a 2D nanomaterial with a sheet-like structure analogous to graphene, in which boron and nitrogen form a hexagonal geometry with a 1:1 atomic ratio. 2D-hBN is typically considered to be an insulating material due to its large band gap (3.6–7.1 eV) [4]. However, the introduction of physical defects into the surfactant free (pristine) form of 2D-hBN can lower the bandgap to ca. 2.36 eV [16]. After applying 2D-hBN on a Screen-Printed electrode (SPE) surface, Khan et al. [17] found that the overpotential of ORR in 0.1 M H_2_SO_4_ lowered to 280 mV (vs. a saturated calomel electrode (SCE)), compared to 500 mV at a bare/unmodified SPE. The ORR over-potentials of 2D-hBN/SPEs and bare/unmodified SPE were ca. 280 mV (vs. a saturated calomel electrode (SCE)) and 500 mV (vs. SCE), respectively. The lower ORR overpotential exhibited by 2D-hBN/SPEs was attributed to the interaction between the underlying rough graphitic SPE surface and the edge sites of 2D-hBN [17,18]. Previous studies have demonstrated that the band gap of 2D-hBN can be transformed from that of an insulator to that of a semiconductor when its structure is doped (via a series of hydrogenation and oxygen passivation of the boron and/or nitrogen at the edge sites) [17]. It has also been reported the synergistic effect between carbon and hBN nanosheets resulted in an affinity for O_2_ adsorption, in addition to high spin and charge density [19]. Supported by density-functional theory (DFT) and experimental investigations, Chen et al. [20]. achieved the production of H_2_O_2_ via the 2e^−^ ORR pathway at the interface between h-BN domains and graphitic lattices [20]. It is also important to note that the electrocatalytic behaviour of 2D-hBN needs further research efforts in order to fully elucidate and comprehend its capabilities towards electrochemical applications.

A literature review pertaining to 2D-hBN electrocatalysts that have been utilised towards ORR is shown in Table 1, which shows that drop-casting and spin-coating methods are often the chosen manufacturing method. These are widely known as not fit for large scale manufacturing; therefore, Table 1 is evidencing the lack of mass-producible manufacturing methods towards the use of 2D-hBN in electrochemical applications. It is clear that there is a repeated use of drop-casting as a deposition method for 2D nanomaterials. Drop-casting surface modification methodologies involve the immobilisation of an aliquot of a suspension (containing the electrocatalyst suspended in a solvent) to the electrode surface, allowing the solution to evaporate and thereby leaving the electrocatalyst electrically wired to the electrode surface. Whilst this is a simple technique to generate an electrochemical signal output at a given material, it has numerous drawbacks, including poor reproducibility, time consuming, uncontrollable distribution of the electrocatalyst and vitality [21,22], thereby limiting their commercialisation. For example, the drop-casting method requires the addition of a few μL of modifier at a time on each working electrode, allowing it to dry, followed by successive additions if necessary. Such a procedure has inevitable made drop-casting a time-consuming method. Even an experienced analyst would likely require a day to manufacture a modified electrode using drop-casting. On the contrary, bulk-modified electrodes would allow the manufacture/printing of a tremendously high number of electrodes each day. These drawbacks are evident when utilising 2D nanomaterials that display different electrochemical signals dependent on whether they are in their monolayer, quasi or bulk forms [5,23,24].

Herein, we report a facile and scalable methodology for the production of SPEs containing varying masses of 2D-hBN incorporated in their bulk (2D-hBN/SPEs). The ORR catalysis of 2D-hBN/SPEs is assessed with regards to their onset potential, achievable current densities and reaction mechanism in acidic conditions. The fabrication technique utilised herein has the potential to replace the less-reproducible and time-consuming drop casting technique of 2D-hBN, offering a rapid, repeatable and ready-to-use alternative. This approach enables the mass production of bespoke electrode platforms that can be employed towards a range of applications.

## 2. Experimental Section

### 2.1. Chemicals

All chemicals utilised were of analytical grade and were used as received from Sigma-Aldrich without any further purification. This includes the 2D-hBN utilised throughout this work. All solutions were prepared with deionised water of resistivity not less than 18.2 MΩ cm^−1^ and (when necessary) were vigorously degassed with high purity, oxygen free nitrogen prior to electrochemical measurements.

### 2.2. Production of 2D-hBN/SPEs

A screen-printing process was utilised to fabricate 2D-hBN/SPEs, as depicted in Figure 1. This technique has been previously reported for the in-house fabrication of graphitic SPEs [30,31]. As illustrated in Figure 1A, SPEs consist of a graphite working macroelectrode (3.1 mm diameter) and a graphite counter electrode, both of which are printed using carbon–graphite ink (product code: C2000802P2; Gwent Electronic Materials Ltd., Derbyshire, UK). An Ag/AgCl reference electrode produced using an Ag/AgCl ink (product code C2040308D2; Gwent Electronic Materials Ltd., Derbyshire, UK) and dielectric ink (product code: D2070423D5; Gwent Electronic Materials Ltd., Derbyshire, UK) constitute the second and third printed electrode layers, respectively. All inks were mechanically activated by using an overhead stirrer (Caframo^TM^ Petite) at 900 rpms, for 30 min prior to their use to ensure their homogeneity. The inks were sequentially screen-printed onto the substrate (Autostat, 250 µm thickness) with each ink layer being cured within a dry oven (60 °C) for 30 min before the next ink layer was applied (see Figure 1). The screen-printing process (performed using a microDEK1760RS screen-printing instrument (DEK, Weymouth, UK) involves a flood bar flooding a porous mesh with ink, this ink was then forced/pressed through the porous stencil sections of the mesh onto the substrate by a squeegee (See Figure 1B). Our printing method deposits a ca. 8 μm layer of ink on each step. After the carbon, reference and dielectric layers have been applied and cured, and the SPE is ready for use. In order to modify the SPEs with 2D-hBN, 2D-hBN was dissolved using ethanol (99.5%) and drop-casted onto the working area of the SPE, the ethanol subsequently evaporates, leaving a randomly distributed coverage of 2D-hBN upon the SPE surface (see Figure 1C). Alternatively, the 2D-hBN was incorporated into a bespoke screen-printable ink, resulting in a controlled and even distribution of 2D-hBN upon the SPE surface (see Figure 1A). This method involves 2D-hBN being incorporated into the graphitic ink at a desirable percentage of the mass of particulate (MP) and the mass of carbon-graphite ink formulation (MI) utilised in the printing process. Both MP and MI typically vary over the 0–15% range. As 2D-hBN was incorporated into the ink, its viscosity would increase and additional solvent was required. Otherwise, when the percentage reached 15%, it became too viscous to reliably print using the screen-printing technique utilised herein. For the fabrication of 2D-hBN/SPEs, 2D-hBN was screen-printed on top of the working electrode and cured (60 °C for 30 min) [32]; the 2D-hBN/SPEs were then ready for use.

### 2.3. Electrochemical Measurements

Voltammetric measurements were performed using an ‘Autolab PGSTAT 101’ (Metrohm Autolab, Utrecht, The Netherlands) potentiostat. All measurements were conducted using a conventional three electrode system. The working electrodes were the 2D-hBN/SPEs or a benchmark 3 mm diameter Pt working electrode (BAS, USAP). The in-built SPE’s Ag/AgCl reference and carbon counter electrodes were removed and replaced with an external SCE reference (warning: this electrode must be carefully handled owing to the health risks associated with mercury vapour from the electrode) and Ni mesh, respectively. In all experimental procedures, 0.1 M H_2_SO_4_ (99.999%, doubly distilled for trace metal analysis) was used. In our work, 0.9 mM electrolyte solution (100 mL) was oxygenated by subjecting it to rigorous bubbling of 100% medicinal grade oxygen for 45 min [28,29]. Note that the ORR onset potential was recorded at the potential when the current deviates from the background current by 25 µA cm^−2^. Measurements were performed using three electrode samples (*n* = 3; also note that we have previously reported that the intra- and inter-batch reproducibility of our electrodes is around 5%).

### 2.4. Physicochemical Characterization Equipment

Transmission electron micrographs (TEMs) were obtained using a JEM-2100 coupled with an OI Aztec 80 mm X-max EDS detector under conventional bright-field conditions. A commercially sourced 2D-hBN sample was placed onto a holey carbon film supported on a 300 mesh Cu TEM grid. Scanning electron micrographs (SEMs) and surface element analysis were obtained using a JEOL JSM-5600LV model equipped with an energy-dispersive X-ray spectroscopy (EDS) for the EDS microanalysis. For Raman spectroscopy and X-ray powder diffraction (XRD) analysis, 2D-hBN was placed on a supporting silicon wafer. Raman spectroscopy was performed using a ‘Renishaw InVia’ spectrometer with a confocal microscope (×50 objective) and an argon laser (514.3 nm excitation) at a very low laser power level (0.8 mW) to avoid any heating effects. Spectra were recorded using a 10 s exposure time for 3 accumulations. XRD was performed using a X’Pert powder diffractometer (PANalytical) with a copper source of *K*_α_ radiation of 1.54 Å and *K*_ß_ radiation of 1.39 Å, using a thin sheet of nickel with an absorption edge of 1.49 Å to absorb *K*_ß_ radiation.

X-ray photoelectron spectroscopy (XPS) was used to analyse the commercially procured 2D-hBN. XPS data were acquired using a bespoke ultra-high vacuum system fitted with a Specs GmbH Focus 500 monochromated Al Kα X-ray source; Specs GmbH PHOIBOS 150 mm mean radius hemispherical analyser with 9-channel tron detection; and a Specs GmbH FG20 charge neutralizing electron gun. Survey spectra was acquired over the binding energy range 1100–0 eV using a pass energy of 50 eV, and high-resolution scans were made over the C 1s and O 1s lines using a pass energy of 20 eV. Under these conditions, the full width at half maximum of the Ag 3d_5/2_ reference line is ca. 0.7 eV. In each case, the analysis was an area-average over a region approximately 1.4 mm in diameter on the sample surface, using the 7 mm diameter aperture and lens magnification of ×5. The energy scale of the instrument was calibrated according to ISO standard 15472, and the intensity scale was calibrated using an in-house method traceable to the UK National Physical Laboratory. Finally, data were quantified using Scofield photoelectric cross sections corrected for the energy dependencies of the electron attenuation lengths and instrument transmission. Data interpretation was carried out using CasaXPS software v2.3.16.

## 3. Results and Discussion

### 3.1. Physicochemical Characterisation of 2D-hBN

It was important to assess the quality and purity of 2D-hBN utilised within this study; therefore, a full physicochemical characterisation was performed. This involved Raman spectroscopy, transmission electron micrographs (TEMs), scanning electron micrographs (SEMs), SEM/energy dispersive X-ray spectroscopy (EDS), X-ray powder diffraction (XRD) and X-ray photoelectron spectroscopy (XPS).

ESI Appendix A shows a typical Raman spectrum of the 2D-hBN, which indicates a characteristic peak at 1366 cm^−1^ due to the E_2g_ phonon mode [33,34]. The transmission electron micrograph shown in ESI. Appendix A indicates that 2D-hBN has an average particle size (lateral) of between 50 and 250 nm, in agreement <250 nm reported by the commercial manufacturer [35]. XRD was performed to assess the crystallinity of the 2D-hBN. ESI Appendix A shows the characteristic diffraction peaks at 2θ = 26.70°, 41.56° and 44.38°, which are accordingly indexed to the (002), (100) and (101) single crystal faces within 2D-hBN [36,37]. SEM and EDS of the pristine 2D-hBN samples are shown in ESI Appendix A, confirming the presence of B and N elements from the powders utilised herein. XPS was performed to assess the surface elemental composition of 2D-hBN. ESI Appendix A is the XPS spectra showing the presence of a single component at 190.8 eV in the B 1s spectrum. The main peak at 398.4 eV corresponds to the N 1s spectrum; as previously shown in other literature [38]. The stoichiometry of 1:1 for B: N is noted and the binding energies for the B 1s and N 1s photoelectron peaks agree well with the expected values for 2D-hBN. The characterisation results above show 2D-hBN utilised within this study to be of high quality and contaminant-free (pristine).

### 3.2. Electrochemical Performance of the 2D-hBN/SPEs towards the ORR

Once the commercially sourced 2D-hBN had undergone a full physicochemical characterisation, it was utilised to produce bespoke screen-printable inks that were subsequently used to produce 2D-hBN incorporated screen-printed electrodes (2D-hBN/SPEs). 2D-hBN/SPEs with 1, 5, 10 and 15% mass incorporations of 2D-hBN were produced and electrochemically explored towards the ORR in an acidic solution of 0.1 M H_2_SO_4_.

The fundamental electron transfer properties of ORR catalysis at 2D-hBN drop-casted SPEs have previously been reported [16,17,18]; however, drop-casting methos are not considered a scalable deposition technique [16,18,39]. Therefore, the mass-produced 2D-hBN/SPEs manufactured by using an automated screen-printed instrument should mitigate issues such as unregulated and non-continuous location and dispersion of BN clusters [40,41].

Initially, it was important to benchmark the ORR activity of an unmodified/bare SPE and Pt electrode against the range of SPEs produced herein in 0.1 M H_2_SO_4_. Figure 2 depicts the linear sweep voltammograms (LSVs) obtained at the unmodified/bare SPE, 1, 5, 10 and 15% 2D-hBN/SPE, which correspondingly exhibit ORR onset potential values of −1.09, + 0.05, −0.94, −0.89, −0.96 V and −1.03 V (vs. SCE). The achievable peak currents of –1.10, −2.26, −1.65, −1.71, −1.74 and −2.02 mA cm^−2^ are accordingly exhibited by the unmodified/bare SPE, Pt electrode (see ESI Appendix A), 1, 5, 10 and 15% 2D-hBN/SPE. When the onset potential is plotted against the mass percentage of 2D-hBN, as depicted in Figure 3A, a general increasing relation is obtained between 0 and 5%, before the onset potential decreases at a higher mass percentage. Therefore, there is a critical mass of 2D-hBN of ca. 5% where 2D-hBN/SPE exhibited the least electronegative ORR onset potential. The optimal ORR catalysis is displayed by 5% 2D-hBN/SPE; this is a likely a result of 2D-hBN being electrically wired to the underlying graphitic substrate whilst also being exposed to the electrolyte. In contrast, at a greater mass of modification, there is significant shielding of the majority of 2D-hBN by the uppermost layer of 2D-hBN, thus leading to decreased affinity to O_2_ adsorption [19,42]. The decline in ORR catalysis exhibited by 10% and 15% 2D-hBN/SPE can be attributed to the agglomeration of 2D-hBN upon the SPE’s surface where bulk coagulation leads to a blocking of active sites that exhibit affinity for O_2_ and H^+^ ions within the electrolyte [32,43]. Figure 3B shows that as the percentage incorporation of 2D-hBN increases within 2D-hBN/SPEs, there is an increase in the achievable current. The recorded peak current density of –2.02 mA cm^−2^ exhibited by 15% 2D-hBN/SPE corresponds to a signal increase of 220% compared to the bare/unmodified SPE’s response. It is important to note that these responses are also higher than ca. 25 µA at an SPE drop-casted with 324 ng of 2D [14]. The observed increase in peak height is likely a result of a higher mass of conductive 2D-hBN, comparative to the mass of graphite across the range of SPEs.

To explore this further, scanning electron micrographs of 2D-hBN/SPEs at various percentage 2D-hBN incorporations (1–15%) were obtained. Figure 4A illustrates that small amounts of 2D-hBN are visible as layered disk-like stacked vertical heterostructures (ca. 200 nm in size) within 1% 2D-hBN/SPEs. Figure 4B–D correspondingly show a clear increase in the presence of 2D-hBN located upon the graphitic sites of the 5, 10 and 15% 2D-hBN/SPEs. There is substantial agglomeration of 2D-hBN upon the surface of the 10% 2D-hBN/SPE and 15% 2D-hBN/SPE, with an apparent size increase from ca. 200 nm to 500 nm of the 2D-hBN discs. Thus, as stated previously, the ORR kinetics of 2D-hBN/SPEs is hindered at higher 2D-hBN masses, which may be due to the observed agglomeration of 2D-hBN causing blockage of the active sites. However, it is clear that novel 2D-hBN inks formulated at lower percentages and offer optimal ORR catalysis within 2D-hBN/SPEs.

The reproducibility of 2D-hBN/SPEs was measured by determining the percentage relative standard deviation (% RSD, *n* = 3) in their achievable ORR peak currents (based on LSVs at 100 mV s^−1^). The bare/unmodified SPE exhibited a %RSD of 4.30%. The 1, 5, 10, and 15% 2D-hBN/SPEs (*n* = 3; 100 mV s^−1^) correspondingly produced % RSD values of 3.03, 3.23, 3.82 and 3.46 %. These values are significantly lower than those previously reported for SPEs, where 2D-hBN was drop-cast upon their surface (8.86 and 12.59% for 108 and 342 ng of 2D-hBN, respectively) [33]. Thus, in comparison to the drop-casting technique, in which the % RSD values are significantly affected upon increasing the immobilisation of any 2D-hBN material due to the instability of the modified layer (pristine or surfactant exfoliated), 2D-hBN-impregnated SPEs offer consistently low % RSD values at the varied 2D-hBN incorporations.

### 3.3. Evaluation of the ORR Mechanism

2D-hBN/SPEs demonstrated greater electrocatalytic activity, as evidenced by the changes in voltammetric signals than the unmodified/bare SPE. Since the electrochemical reduction of oxygen can follow the two-step two-electron peroxide pathway or the direct four-electron pathway, as described in the introduction, it is important to study the role of 2D-hBN in each respective mechanism.

In order to estimate the selectivity of 2D-hBN to favor either the two or four electrochemical ORR pathways, Tafel analysis was performed for unmodified and 2D-hBN/SPEs variants. A plot of ‘ln current (*I*)’ vs. ‘*E*_p_’ was constructed from an analysis of voltammograms corresponding to the electrochemical reduction of oxygen using Equation (4).
(4)$$\frac{\delta \ln\;I}{\delta Ep}=\frac{\left(\alpha n^{\prime }\right)F}{RT}$$

*I* denotes current, *Ep* is peak potential *α* is the electron transfer coefficient, *F* is the Faraday constant and *n*′ is the number of electrons transferred in the rate determining step. *R* is the universal gas constant, and *T* is the absolute temperature. The slope of δln I against δEp will yield $\left(\alpha n^{\prime }\right)$. Using this value and the following Randles–Ševćik Equation (5) for an irreversible electrochemical process, the number of electrons transferred overall, *n*, was deduced [44,45].
(5)$$I_{p}=-0.496\;{\left(\alpha n\prime \right)}^{\frac{1}{2}}\;nFAC{\left(nFvD/RT\right)}^{\frac{1}{2}}$$

*A* is the electrode area; *C* is bulk the concentration of oxygen within the analyte under investigation (where in this case, a fully saturated acidic solution is reported to correspond to 0.9 mM [46,47]); v is the scan rate; and *D* is the diffusion coefficient of O_2_, which has been reported to be 2.0 × 10^−5^ cm^2^ s^−1^. Equation (5) calculates “*n*” by introducing the recorded peak current “*Ip*” and other corresponding values (αn′ from Equation (4)). Consequently, for the unmodified SPEs and 1, 5, 10 and 15% 2D-hBN/SPEs, in all cases, *n* was calculated to be no greater than 2 and, therefore, indicate that the reduction of oxygen can follow the two-step two-electron peroxide pathway. These values confirm that the electrochemical reduction of oxygen using unmodified and 2D-hBN/SPEs preferentially drives the production of hydrogen peroxide (H_2_O_2_). Note that our previous work demonstrated that pristine 2D-hBN and surfactant exfoliated 2D-hBN followed the same two-electron pathway and produced H_2_O_2_. Furthermore, previous studies by Uosaki et al. [4] reported that 2D-hBN exhibited selectivity for the two-electron pathway at a gold substrate, and Jaramillo et al. [20] also reported that the interface of *h*-BN domains and graphene creates unique carbon-hBN domains that preferentially select a two-electron pathway for ORR. This is due to the material binding oxygen weakly, which tends to favour H_2_O_2_ production rather than H_2_O. Jaramillo et al. [20] supports this assertation via the employment of a density functional theory.

H_2_O_2_ yields were estimated for unmodified and 2D-hBN/SPEs. First, the capacitance of an electrochemical process is estimated utilizing the following: C=I/v, where *C* is the capacitance, *I* is the observed peak current at a certain potential and *v* is the scan rate. Next, charge (Q) is calculated using the following: Q=CV, where *V* is the potential. Resultantly, Q=nFN, and the number of moles of oxygen electrolysed, N, was readily evaluated. This enables the amount of oxygen electrolysed in the reaction to be calculated. Based on the first step of the two-electron reactions shown in the Section 1, there is a 1:1 stoichiometric ratio between oxygen electrolysed to H_2_O_2_ produced in an electrochemical reaction. Therefore, the estimated H_2_O_2_ yields were calculated for an oxygenated 0.1 M H_2_SO_4_ solution at a fixed volume of 10 mL utilising unmodified and 2D-hBN/SPEs of percentages ranging from 1 to 15% at a scan rate of 100 mV s^−1^. It was estimated that the concentration of H_2_O_2_ electrolysed when utilising unmodified SPEs was 2.51 nM, whereas 2D-hBN-SPEs resulted in the electrolysis of 8.74, 4.89, 4.72 and 5.95 nM of H_2_O_2_ for 1, 5, 10 and 15% h-BN. It is evident that there is an increase in the quantity of H_2_O_2_ produced when utilising 2D-hBN/SPEs for electrochemical oxygen reduction in acidic conditions.

The results presented herein illustrate the incorporation of 2D-hBN within graphitic printable inks to fabricate bulk 2D-hBN/SPEs offering a beneficial, scalable and reproducible alternative approach towards ORR in comparison to the current drop-casting technique. 2D-hBN/SPEs offer a significant improvement in ORR onset potential in comparison to the drop-casting technique, whilst significantly enhancing the achievable current. We have demonstrated that the 2D-hBN bulk-modified SPEs facilitate the in situ generation of H_2_O_2_, which provides a scalable fabrication approach; this is the focus of future work.

## 4. Conclusions

We have demonstrated a facile fabrication technique to produce bespoke 2D-hBN electrocatalytic inks. The subsequent 2D-hBN/SPEs exhibit proficient ORR catalysis in comparison to an unmodified graphitic SPE, with an ORR onset potential lowered by 200 mV at a 5% 2D-hBN/SPE compared to a bare SPE. The electronegative shift in ORR onset potential is less significant than previously reported in pristine 2D-hBN-modified SPEs produced using drop-casting methodology. However, the achievable current of the 2D-hBN/SPEs significantly improved by up to 220% compared to an unmodified SPE. The results described above report a novel 2D-hBN/SPE fabrication approach, which offers a portable and low-cost electrode platform, which can be manufactured by a facile methodology. It is clear that the 2D-hBN/SPEs used within this study exhibit selectivity for the two-electron O_2_ reduction pathway, suggesting that our future work should be focussed on utilising electrode platforms for the in situ generation of hydrogen peroxide as they could be of great commercial interest to the field of wastewater management.

## Figures and Tables

**Figure 1 sensors-22-03330-f001:**
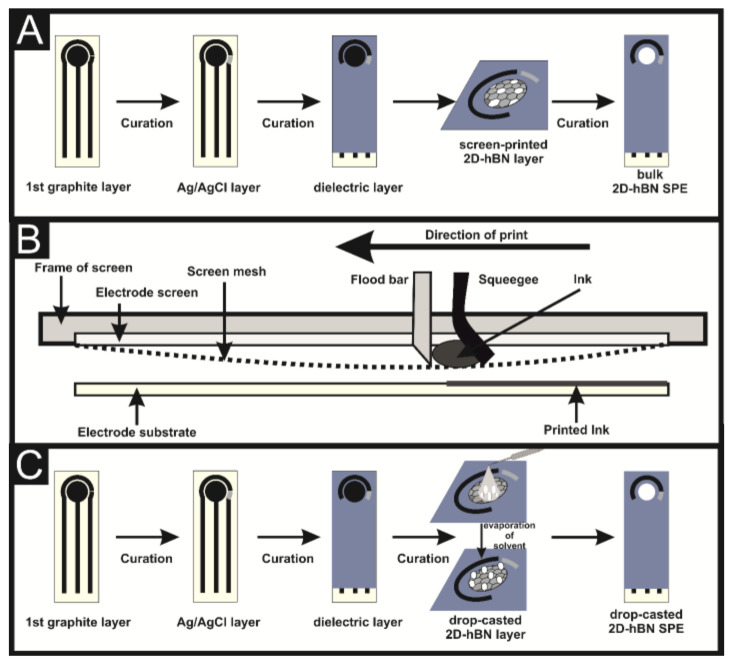
A schematic representation of the screen-printing process utilised to produce the bulk 2D-hBN/SPEs reported herein (**A**), a schematic cross-section of a screen-print (**B**) and a comparison with the usual drop-casting method (**C**).

**Figure 2 sensors-22-03330-f002:**
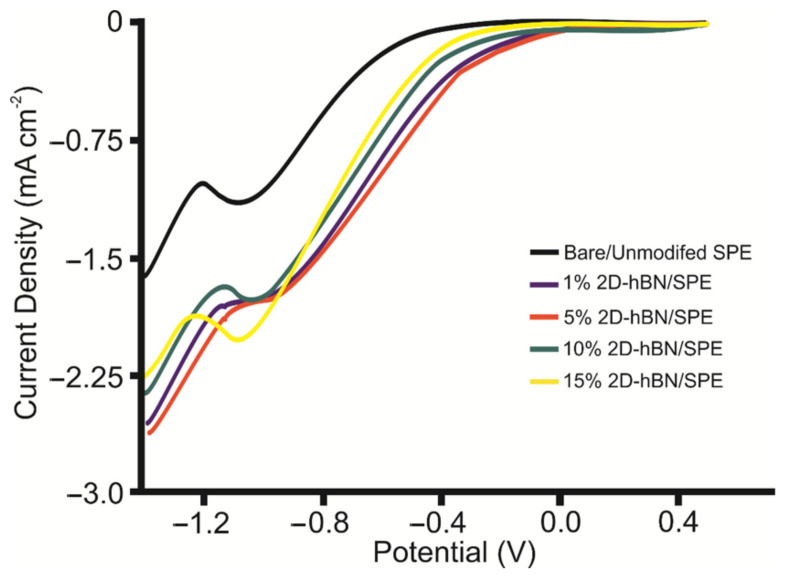
Typical LSVs recorded in an oxygen saturated 0.1 M H_2_SO_4_ solution using unmodified (black line), 1% 2D-hBN (blue line), 5% 2D-hBN (red line), 10% 2D-hBN (green line) and 15% 2D-hBN (yellow line) incorporated SPEs. Scan rate: 100 mV s^−1^ (vs. SCE).

**Figure 3 sensors-22-03330-f003:**
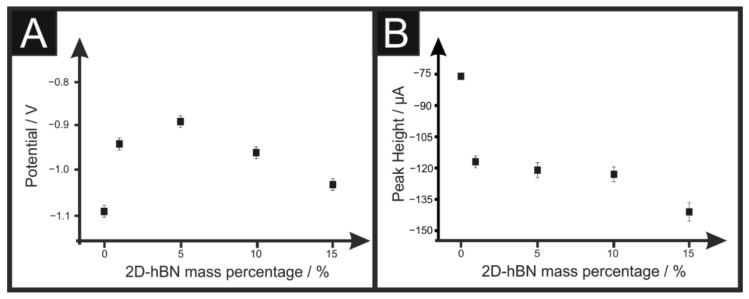
Analysis of voltammograms obtained in an oxygen saturated 0.1 M H_2_SO_4_ solution in the form of a plot of oxygen reduction peak potential vs. incorporated 2D-hBN % (**A**), peak potential vs. mass of pristine 2D-hBN/SPEs and (**B**) peak current vs. mass of pristine 2D-hBN/SPEs, recorded in oxygen saturated 0.1 M H_2_SO_4_. Scan rate: 100 mV s^−1^ (vs. SCE); *n* = 3.

**Figure 4 sensors-22-03330-f004:**
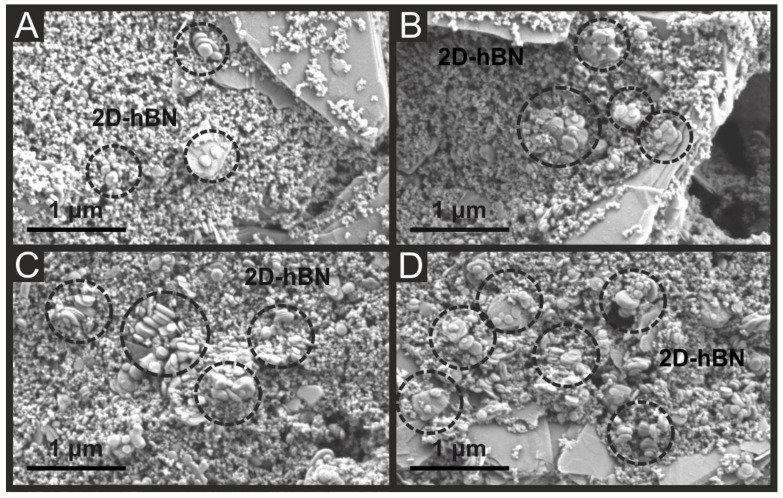
Typical scanning electron micrographs for SPE with 1% 2D-hBN (**A**), 5% 2D-hBN (**B**), 10% 2D-hBN (**C**) and 15% 2D-hBN (**D**). 2D-hBN platelets are evident as layered disk-like stacked vertical heterostructures of ca. 200 nm in size.

**Table 1 sensors-22-03330-t001:** Comparison of 2D-hBN electrocatalysts that have been utilised towards ORR.

Catalyst	Substrate	Loading (mg cm^−2^)	Deposition Method	ORR Potential Onset (mV)	Electrolyte	Reference
*-*	GC	-	-	−780 (vs. SCE)	0.1 M H_2_SO_4_	[17]
*-*	SPE	-	-	−1090 (vs. SCE)	0.1 M H_2_SO_4_	This work
2D-hBN	GC	0.0046	Drop-casted	−1000 (vs. SCE)	0.1 M H_2_SO_4_	[17]
2D-hBN	SPE	0.0046	Drop-casted	−810 (vs. SCE)	0.1 M H_2_SO_4_	[17]
C-doped BN	GC RDE	0.51	Spin-coated	−800 (vs. RHE)	0.1 M KOH	[20]
2D-hBN/CVD Graphene	GC RDE	0.025	Drop-casted	−780 (vs. RHE)	0.1 M KOH	[25]
AuNPs-BNNS	Au RDE	1.26	Drop-casted	−670 (vs. RHE)	0.05 M KOH	[26]
BCN	GC RDE	0.3	Drop-casted	−840 (vs. RHE)	0.1 M HClO_4_	[27]
Surfactant exfoliated 2D-hBN	SPE	0.00053	Drop-casted	−590 (vs. RHE)	0.1 M H_2_SO_4_	[18]
BCN-doped CNTs	GC RDE	0.1	Drop-casted	−920 (vs. RHE)	0.1 M KOH	[28]
BCN nanosheet	GC RDE	1.265	Drop-casted	−650 (vs. RHE)	0.1 M KOH	[29]
2D-hBN	SPE	5%	Screen-printed	−890 (vs. SCE)	0.1 M H_2_SO_4_	This work

Key: GC, glassy carbon electrode; SCE, saturated calomel electrode; SPE, screen-printed electrode; hBN, hexagonal boron nitride; CVD, chemical vapour deposition; AuNPs-BNNS, boron nitride nanosheets decorated with small gold nanoparticles; RDE, rotating disk electrode; BCN, porous boron carbon nitride nanosheets; CNT, carbon nanotubes.

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
