# Peer review of "2D-Hexagonal Boron Nitride Screen-Printed Bulk-Modified Electrochemical Platforms Explored towards Oxygen Reduction Reactions"

_sensors, 2022, doi:10.3390/s22093330_

Round 1
Reviewer 1 Report
In this manuscript, the Authors described various analyses on screen-printed 2D hexagonal boron nitride for oxygen reduction reaction and elucidated its effect on large-scale manufacturing. The Reviewer was a little bit surprised at the fact that no one has ever attempted to employ the screen printing for the pattern formation; someone should easily come into mind to use, since this printing is very widely used for electronic device manufacturing in the world. Anyway, the content seems well organized and has a potential to be published in Sensors. However, the Reviewer thinks that two portions should be revised before publication.
(i) Fig. 4 & P. 5, LL. 213−216
The Authors mentioned “Figure 4(B), (C) and (D) show a clear increase in the presence of 2D-hBN located upon the graphitic sites of the 5, 10 and 15% 2D-hBN/SPEs, respectively. There is substantial agglomeration of 2D-hBN upon the surface of the 10% 2D-hBN/SPE and 15% 2D-hBN/SPE, with an apparent size increase from ca. 200 nm to 500 nm of the 2D-hBN discs.” However, it is very difficult to distinguish from 2D-hBN/SPE and other materials from these images. For example, the size of the disk-like thing indicated by the arrow in Fig. 4(D) seems ~200 nm, smaller than 500 nm. It seems helpful to indicate the 2D-hBM using not arrows but other things such as a circle.
(ii) P. 2−3, LL. 94−96
The Authors mentioned “The fabrication technique utilised herein has the potential to replace the less-reproducible and time-consuming drop casting technique of 2D-hBN, offering a rapid, repeatable and ready-to-use alternative.” It should be helpful information how fast fabrication will be realized if we employ screen printing for large-scale manufacturing. For example, “It takes ** s and ** min to form 100 sensor patterns using screen printing and drop-casting, respectively.” Although the Authors have already mentioned good repeatability (P. 5, LL. 222−224) and high achievable current (P. 5, LL. 203−204) compared to the drop-casting technique, the information about manufacturing time (takt time) should also be important to advocate the priority of the proposed process.
Author Response
Firstly, we would like to thank the reviewer for taking time to assess the manuscript. We have made numerous alterations within the manuscript in order to improve its quality and make it appropriate for publication with Sensors MDPI. We have addressed each of the reviewer’s comments in turn (see blue text below) and highlighted major alterations within the manuscript in yellow, which result in a much better improved version of our manuscript.
Reviewer 1:
Comments and Suggestions for Authors
In this manuscript, the Authors described various analyses on screen-printed 2D hexagonal boron nitride for oxygen reduction reaction and elucidated its effect on large-scale manufacturing. The Reviewer was a little bit surprised at the fact that no one has ever attempted to employ the screen printing for the pattern formation; someone should easily come into mind to use, since this printing is very widely used for electronic device manufacturing in the world. Anyway, the content seems well organized and has a potential to be published in Sensors. However, the Reviewer thinks that two portions should be revised before publication.
We thank the reviewer for their helpful comment above, appreciating the novelty of our manuscript. We can also confirm we have addressed both portions suggested by the reviewer which are addressed below.
(i) Fig. 4 & P. 5, LL. 213−216
The Authors mentioned “Figure 4(B), (C) and (D) show a clear increase in the presence of 2D-hBN located upon the graphitic sites of the 5, 10 and 15% 2D-hBN/SPEs, respectively. There is substantial agglomeration of 2D-hBN upon the surface of the 10% 2D-hBN/SPE and 15% 2D-hBN/SPE, with an apparent size increase from ca. 200 nm to 500 nm of the 2D-hBN discs.” However, it is very difficult to distinguish from 2D-hBN/SPE and other materials from these images. For example, the size of the disk-like thing indicated by the arrow in Fig. 4(D) seems ~200 nm, smaller than 500 nm. It seems helpful to indicate the 2D-hBM using not arrows but other things such as a circle.
We thank the reviewer again, and we can confirm we have updated now Figure 4 with their suggestion. We appreciate it the suggestion and agree that makes easier the boron nitride identification in the SEM images.
(ii) P. 2−3, LL. 94−96
The Authors mentioned “The fabrication technique utilised herein has the potential to replace the less-reproducible and time-consuming drop casting technique of 2D-hBN, offering a rapid, repeatable and ready-to-use alternative.” It should be helpful information how fast fabrication will be realized if we employ screen printing for large-scale manufacturing. For example, “It takes ** s and ** min to form 100 sensor patterns using screen printing and drop-casting, respectively.” Although the Authors have already mentioned good repeatability (P. 5, LL. 222−224) and high achievable current (P. 5, LL. 203−204) compared to the drop-casting technique, the information about manufacturing time (takt time) should also be important to advocate the priority of the proposed process.
We appreciate the reviewer’s comment, and we agree indeed that manufacturing time is a very important parameter when developing large-scale manufacturing methods. We can confirm we have now included this in the manuscript.
Reviewer 2 Report
This article deals with screen-printed platforms dedicated to oxygen reduction reaction (so called ORR). Metals free 2D-hexagonal Boron Nitride are proposed as metal-free electrocatalysts and are incorporated in screen-printed graphitic electrodes which ar part of the platforms. The manufacturing of such platforms by low-cost printing technology is shown as well as the different properties of the catalysts (physicochemical characterization and electrochemical measurements). The paper is clear and well written. Please find below my comments/suggestions which could help to proceed your revisions.
Abstract: even if SPE is defined in the keywords, I suggest to give the definition of this acronym in the abstract
Equation 3 page 2 : in the reaction, shouldn’t it be 2 times H2O2 for the equilibrium of the chemical equation ? .
Shouldn’t the acronym SCE appearing in page 2 be explained as the other acronym?
Paragraph 2.2 , screen-printing process :
Do you have a good adhesion of the electrodes on Teflon?
The choice of screens is important because it will affect the definition of the pattern and the thickness of the layers. COuld you give more details (meshs, emulsion thickness)?
You speak about incorporation of the 2D-hBN in the graphite ink, but could you detail its incorporation. The homogeneity of the paste is important for the reproducibility, and this point should be detailed.
As the amount of 2D-hBN in the ink increases, the viscosity of the paste increases. This will inevitably impact the final thickness. It seems to me important to give the values of screen printed thickness
Do you think that the thickness of the layer also has an impact on the catalysis phenomena? Platforms using drop casting technology were maybe more performant because of a volume difference? What is your opinion.
Have you tested different thicknesses with the same amount of catalyst? It could be relevant to add that information. Also, as the author also worked on drop-casting, it might be relevant to compare the final cured volumes deposited
Figure 3 : N= 3; is it for 3 samples, or three measurements? It is important to give this information
Author Response
Firstly, we would like to thank the reviewer for taking time to assess the manuscript. We have made numerous alterations within the manuscript in order to improve its quality and make it appropriate for publication with Sensors MDPI. We have addressed each of the reviewer’s comments in turn (see blue text below) and highlighted major alterations within the manuscript in yellow, which result in a much better improved version of our manuscript.
Reviewer 2:
Comments and Suggestions for Authors
This article deals with screen-printed platforms dedicated to oxygen reduction reaction (so called ORR). Metals free 2D-hexagonal Boron Nitride are proposed as metal-free electrocatalysts and are incorporated in screen-printed graphitic electrodes which ar part of the platforms. The manufacturing of such platforms by low-cost printing technology is shown as well as the different properties of the catalysts (physicochemical characterization and electrochemical measurements). The paper is clear and well written. Please find below my comments/suggestions which could help to proceed your revisions.
Firstly, we would like to thank the reviewer for taking the time to assess it again and for the positive feedback throughout.
Abstract: even if SPE is defined in the keywords, I suggest to give the definition of this acronym in the abstract
We can confirm we have changed this now.
Equation 3 page 2 : in the reaction, shouldn’t it be 2 times H2O for the equilibrium of the chemical equation ? .
We apologise for the sincere mistake, we agree with the reviewer and we can confirm this has been amended now.
Shouldn’t the acronym SCE appearing in page 2 be explained as the other acronym?
We can confirm we have changed this now.
Paragraph 2.2 , screen-printing process :
Do you have a good adhesion of the electrodes on Teflon?
We are very sorry and we thank the referee for asking this question. We have amended now the manuscript, however we have not used Teflon substrate for this piece of manuscript, but polyester Autostat, 250. mm thickness.
The choice of screens is important because it will affect the definition of the pattern and the thickness of the layers. COuld you give more details (meshs, emulsion thickness)?
We appreciate the reviewer’s interest and we agree on the importance of these parameters. However we have no liberty or capacity to disseminate these values as they are proprietary and under know-how agreements at the moment. We can direct the reviewer to read the following book if interested in the screen-printing process in detail: “Screen-Printing Electrochemical Architectures”; ISBN: 978-3-319-25193-6.
You speak about incorporation of the 2D-hBN in the graphite ink, but could you detail its incorporation. The homogeneity of the paste is important for the reproducibility, and this point should be detailed.
We can confirm we have included now a sentence detailing this on the experimental information.
As the amount of 2D-hBN in the ink increases, the viscosity of the paste increases. This will inevitably impact the final thickness. It seems to me important to give the values of screen printed thickness
We can confirm we have included now a sentence detailing this on the experimental information
Do you think that the thickness of the layer also has an impact on the catalysis phenomena? Platforms using drop casting technology were maybe more performant because of a volume difference? What is your opinion.
That is a very interesting suggestion and we have not performed thickness comparisons within our screen-printing electrode investigations, therefore we are in no position of making a statement about it at this point in time. However if one were to make an educated opinion we could only initially say that, electrochemistry happening at the interface of the surface-electrolyte, the most important parameter would be the resistance of both circuits and solution.
Have you tested different thicknesses with the same amount of catalyst? It could be relevant to add that information. Also, as the author also worked on drop-casting, it might be relevant to compare the final cured volumes deposited
We appreciate the reviewers suggestion, however and as mentioned above, we have not performed thickness comparisons.
Figure 3 : N= 3; is it for 3 samples, or three measurements? It is important to give this information.
We thank the reviewer for their suggestion, we can confirm we have specified this within the experimental section.
Reviewer 3 Report
Please see attached Review Report.

Author Response
Firstly, we would like to thank the reviewer for taking time to assess the manuscript. We have made numerous alterations within the manuscript in order to improve its quality and make it appropriate for publication with Sensors MDPI. We have addressed each of the reviewer’s comments in turn (see blue text below) and highlighted major alterations within the manuscript in yellow, which result in a much better improved version of our manuscript.
Reviewer 3:
We thank the reviewer for their helpful comment above, we apologise for the raised issue and we can confirm we have addressed this throughout all the manuscript to make it more inclusive and friendly to a wider audience.
We thank and we can confirm we have expanded the readers and direct them to appropriate literature which addressed the issue directly.
We can confirm we have updated this now and we have also included a sentenced mentioning that this is a hot topic and the electrocatalytic behaviour of 2D-hBN needs further research efforts, in order to fully elucidate and comprehend its capabilities towards electrochemical applications.
We thank the reviewer, we apologise for the typos and we can confirm we have amended all the relevant changes and suggestions, were appropriate.
Round 2
Reviewer 1 Report
Thank you very much for answering the questions. Although the content of the revised manuscript seems suitable for publication in Sensors, there are some typo. For example, "(p.2) mas-producible", "(p.5) before the next ink layer were applied (see Figure 1)", and "(p.10) endveour". Please check the manuscript thoroughly before the publication.
Author Response
Dear reviewer,
We kindly thank you for taking the time to read and review the manuscript. We appreciate all the feedback give throughout and we can confirm that we have re-checked all the manuscript and corrected when necessary (as marked within the highlighted in yellow). Thank you in particular for those mention typos, we truly apologise and we can confirm they are corrected now.
Reviewer 2 Report
To the authors: I think that the paper can now be published.
Juste one last comment on the substrate. Is it thickness of 250mm, or 0.250mm?
Author Response
We thank the reviewer again for the positive opinion and feedback give during all this reviewing period. We also want to thank the reviewer for brining this typo on the manuscript which has now been corrected (to 250 µm) for their suggestion, we can confirm we have specified this within the experimental section.
Reviewer 3 Report
Please see attached Review Report.

Author Response
We thank the reviewer for their helpful comments throughout. We can confirm that we have explained within the manuscript about the use of such equations. We apologise for the raised issue, and we can only say that all of the sincere errors that the authors have detected, have been corrected, however we believe that there is a proofs stage within the publication process for these kind of correction.
We thank the reviewer, we apologise for all the mistakes and we can confirm we have amended all the relevant changes and expand upon suggestions, were appropriate.
The reply have been attached, please check.

Round 3
Reviewer 3 Report
Please see attached Review Report.

Author Response
Thanks for your comments, please find the reply on the attachment.
